# COVID-19 Severity and Mortality after Vaccination against SARS-CoV-2 in Central Greece

**DOI:** 10.3390/jpm12091423

**Published:** 2022-08-31

**Authors:** Athina A. Samara, Stylianos Boutlas, Michel B. Janho, Konstantinos I. Gourgoulianis, Sotirios Sotiriou

**Affiliations:** 1Department of Embryology, Faculty of Medicine, School of Health Sciences, University of Thessaly, 41110 Larissa, Greece; 2Department of Respiratory Medicine, Faculty of Medicine, University of Thessaly, 41110 Larissa, Greece

**Keywords:** COVID-19, vaccination, risk factors, effectiveness, mortality

## Abstract

Background: Vaccination against SARS-CoV-2 (COVID-19) has become crucial for limiting disease transmission and reducing its severity, hospitalizations and mortality; however, despite universal acceptance, vaccine hesitancy is still significant. In the present manuscript, we aim to assess COVID-19-attributed mortality after the prevalence of new variants of the virus (Delta and Omicron viral strains) and to evaluate the vaccination effect. Methods: All patients that were hospitalized due to COVID-19 infection in the Respiratory Department of a tertiary referral center in central Greece between 1st of June 2021 and 1st of February 2022 were included in the present study. Results: 760 consecutive patients were included in the study; 89 (11.7%) were diagnosed with severe COVID-19 and 220 (38.7%) patients were fully vaccinated. In logistic regression, increased age (aOR = 1.12, *p* < 0.001), male gender (aOR = 2.29, *p* = 0.013) and vaccination against SARS-CoV-2 virus (aOR = 0.2, *p* < 0.001) were associated with mortality attributed to COVID-19 with a statistically significant association. Moreover, increased age (aOR = 1.09, *p* < 0.001), male gender (aOR = 1.92, *p* = 0.025) and vaccination against SARS-CoV-2 virus (aOR = 0.25, *p* < 0.001) were statistically significantly associated with clinical severity of COVID-19 infection. However, when comparing the length of hospitalization between vaccinated and unvaccinated patients, the difference was not statistically significant between the two groups (*p* = 0.138). Conclusions: Vaccination against SARS-CoV-2 virus had a protective effect in terms of mortality and clinical severity of COVID-19 during the fourth wave of the pandemic in Central Greece. The national vaccination policy has to focus on vulnerable populations that are expected to benefit the most from the vaccine’s protection.

## 1. Introduction

SARS-CoV-2 is a member of the coronavirus family, a group of enveloped single-stranded RNA viruses. Over the last two years, the SARS-CoV-2 virus (COVID-19) pandemic has spread globally, affecting almost every country worldwide [1]. Clinical signs of COVID-19 differ from mild illness to very severe disease requiring hospitalization with an often fatal outcome [2]. In persons with critical clinical illness due to SARS-CoV-2, the respiratory system is the most commonly affected. However, the virus can affect all systems [3].

The virus binds to angiotensin-converting enzyme 2 (ACE2) receptors present in almost all vascular endothelial cells [4]. COVID-19 mortality and morbidity are affected by many different factors, i.e., gender, aging and several chronic diseases such as chronic respiratory disease/asthma, heart arrhythmias, hypertension, coronary disease, diabetes and neoplasia [5]. Inherited diseases and genetic predispositions such as hemoglobinopathies in homozygous or heterozygous status can also alter disease outcome [6]. 

Vaccination against SARS-CoV-2 has become crucial for limiting disease transmission and reducing its severity, hospitalization and mortality [7]. More specifically, data from systematic reviews underline the efficacy of vaccination in terms of clinical severity and mortality [8,9]. Despite universal acceptance, vaccine hesitancy is still significant [10]. Finally, the prevalence of the Omicron variant changed the data in relation to previous emerging virus strains [11]. In this context, analysis of real-world data regarding the clinical severity and mortality of COVID-19 during the next stages of the pandemic is essential in order to guide the national health policy.

The first Omicron sequence available, however, was from a specimen collected on 11 November 2021 in Botswana. Ever since the identification of Omicron, the variant appears to have rapidly spread. The early doubling time of the Beta, Delta and Omicron variants was calculated to be about 1.7, 1.5 and 1.2 days, respectively [12]. These data indicate that the Omicron variant is probably more infectious than the Delta and Beta variants. Analysis of the genomic sequences of the Omicron variant has revealed a high number of nonsynonymous mutations, including several ones in spike that have been proven to be involved in transmissibility, disease severity and immune escape [13].

In the present manuscript, we aim to assess COVID-19-attributed mortality and clinical severity after the prevalence of new variants of the virus (Delta and Omicron viral strains), to identify common risk factors and to evaluate the vaccination effect in fully vaccinated patients.

## 2. Materials and Methods

### 2.1. Settings

All consecutive patients that were hospitalized due to COVID-19 infection in the Respiratory Department of a tertiary referral center in Central Greece (University Hospital of Larisa) between 1 June 2021 and 1 February 2022 were included in the present study. Final diagnosis was established with a positive SARS-CoV-2 real-time polymerase chain reaction (RT-PCR) molecular test.

### 2.2. Participants and Study Design

A retrospective analysis of medical and laboratory records was conducted. An anonymous database of all patients that needed hospitalization due to COVID-19 infection was created and included both the medical history of common morbidities and clinical records during their hospitalization. Basic demographic characteristics of age and gender and common morbidities, including atrial fibrillation, chronic respiratory disease, coronary disease, diabetes, malignancy and hypertension were recorded. Moreover, vaccination status was recorded from the national vaccination registry.

As primary outcome of the present study was the association of clinical and demographic factors with mortality due to COVID-19 in infected individuals. Furthermore, clinical severity of symptomatology, classified according to the World Health Organization [14], and the length of hospitalization were considered as secondary endpoints. 

Current smoking-status parameter was excluded from multivariate analysis due to insufficient data in the patients’ records, and based on the univariate analysis, no statistically significant correlations between smoking and mortality attributed to COVID-19 were found. Furthermore, patients <18 years of age were excluded from the present study.

### 2.3. Ethical Considerations

Experimental therapeutic protocols were not applicable in this study. All data were analyzed anonymously, using code numbers with respect to the patient’s privacy, and collected in the context of routine diagnostic and therapeutic procedures. The study conformed to the Research and Ethical Committee guidelines of the University Hospital of Larissa.

### 2.4. Statistical Analysis

Analysis was carried out with SPSS version 26.0. Categorical variables are described with the use of frequency and relative frequency. Continuous variables are described with means and standard deviation. The analysis of continuous variables was conducted using the Mann–Whitney U test since the assumption of normal distribution was violated. Data were checked for deviation from normal distribution using Shapiro–Wilk normality test. Categorical data were analyzed with the use of Chi-square test or Fisher’s exact test; multivariate analysis was performed in the form of binary logistic regression. For all analyses, a 5% significance level was set. 

## 3. Results

In total, 760 consecutive patients were hospitalized in a 7-month period in our department and included in the present study. A total of 671 (88.3%) of patients were diagnosed with moderate symptomatology of COVID-19, and 89 (11.7%) experienced severe symptoms. Furthermore, 220 (38.7%) patients were fully vaccinated against SARS-CoV-2 with two doses of one of the four vaccines licensed in Greece (Pfizer/BioNTech Comirnaty, AstraZeneca/AZD1222, Janssen/Ad26.COV 2.S Johnson & Johnson, Moderna COVID-19-mRNA 1273 vaccines). COVID-19 infection was the single most common official cause of death for our study participants, as registered in hospital archives. The fatality rate was calculated at 10.1% (77/760).

In Table 1 are displayed the results of univariate and multivariate statistical analysis regarding mortality risk due to COVID-19 infection. In univariate analysis, increased age (*p* < 0.001), atrial fibrillation (OR = 3.47, *p* < 0.001), chronic respiratory disease (OR = 2.59, *p* = 0.004), coronary disease (OR = 3.83, *p* < 0.001), diabetes type II (OR = 2.5, *p* < 0.001), dyslipidemia (OR = 2.94, *p* < 0.001) and hypertension (OR = 3.41, *p* < 0.001) have been associated with increased mortality attributed to COVID-19. More specifically, regarding the cardiovascular system, patients with atrial fibrillation, coronary disease and hypertension had almost 3.5, 4 and 3.5 times increased mortality risk, respectively, compared with patients without these comorbidities. Moreover, diabetic patients were associated with 2.5 times increased possibility of mortality, while patients with chronic respiratory diseases were 2.6 times more likely to decease after the COVID-19 infection.

In logistic regression of our primary outcome, only increased age (aOR = 1.12, *p* < 0.001), male gender (aOR = 2.29, *p* = 0.013) and vaccination against SARS-CoV-2 virus (aOR = 0.2, *p* < 0.001) were associated with mortality attributed to COVID-19 with a statistically significant association. More precisely, every additional year of age increased the possibility of death by 12% and male patients had almost 2.5 times increased mortality risk compared to females. Moreover, vaccinated patients were associated with a 5 times decreased mortality risk (Figure 1). A result that is worth being underlined is the fact that after the vaccination effect, in multivariate analysis, severe and common comorbidities were not associated with increased mortality risk.

Table 2 displays the results of univariate and multivariate statistical analysis regarding the clinical severity of COVID-19. Increased age (*p* < 0.001), atrial fibrillation (OR = 2.76, *p* < 0.001), chronic respiratory disease (OR = 2.37, *p* = 0.007), coronary disease (OR = 3.01, *p* < 0.001), diabetes type II (OR = 2.3, *p* = 0.002), dyslipidemia (OR = 2.85, *p* < 0.001) and hypertension (OR = 2.91, *p* < 0.001) have been associated with increased mortality attributed to COVID-19. However, similar to the mortality risk assessment, in logistic regression, only increasing age (aOR = 1.09, *p* < 0.001), male gender (aOR = 1.92, *p* = 0.025) and vaccination against SARS-CoV-2 virus (aOR = 0.25, *p* < 0.001) were statistically significantly associated with clinical severity of COVID-19 infection. In this context, every additional year of age increased the risk of severe disease by 9%, while male patients had almost double the risk of severe COVID-19. Similar to the mortality risk, unvaccinated individuals had four times a greater risk of severe infection compared to vaccinated patients (Figure 2).

However, when comparing the length of hospitalization between vaccinated and unvaccinated patients, the difference was not statistically significant between the two groups (*p* = 0.138) (Figure 3).

## 4. Discussion

The present study confirmed the protective effect of vaccination against the SARS-CoV-2 virus in terms of mortality and clinical severity of COVID-19 during the fourth wave of the pandemic in Central Greece. More specifically, vaccinated patients were associated with a 5 times decreased mortality risk and 4 times decreased risk of severe infection compared to unvaccinated individuals. Furthermore, the vaccination coverage between the hospitalized due to COVID-19 infection patient was relatively low at 28.9%, compared with the national vaccination coverage during the same time period. Additionally, in multivariate analysis, except for vaccination, male gender and aging were also identified as independent risk factors for mortality and increased clinical severity of COVID-19 infection.

Identifying mutations of SARS-CoV-2 is crucial, especially those with significant clinical impact and variants of concern (VOC), as they can modify public health policies, surveillance and immunization strategies [15,16]. Delta was the predominant SARS-CoV-2 variant during the fourth COVID-19 wave in many countries worldwide, including Greece; however, the novel Omicron variant rapidly spread due to the increased transmissibility of the variant [17,18]. Early observations suggest that the Omicron outbreak occurred more quickly and with larger magnitude, and despite substantial increases in vaccinations and prior infections, may be less severe than those caused by other virus strains [19,20]. In line with these observations, the fatality rate of the present study was 10.1% compared to the 24.9% that was observed during the second and third waves of the pandemic in Central Greece [6].

Developing COVID-19 vaccines within a short timeframe has raised several concerns about the safety and efficacy of the vaccines, which have been assessed by many studies. A recently published systematic review [21] analyzed the efficacy of different licensed vaccines based on 42 original studies and concluded that COVID-19 vaccines successfully reduced the rates of infections, severity, hospitalization and mortality. More specifically, the Pfizer/BioNTech vaccine was the most extensively studied among the COVID-19 vaccines with >90% effectiveness, followed by the Moderna vaccine with >80% effectiveness against infection, the AstraZeneca vaccine with 80.7% effectiveness against infection after the second dose and 74% effectiveness against infection after the first dose, and a single dose of the Johnson & Johnson vaccine with >60% effectiveness against infection [22]. 

The determinants of effectiveness of the approved vaccine against SARS-CoV-2 virus and breakthrough rates are yet to be determined, especially in light of the emergence of viral VOC [23]. In a recently published prospective study of fully vaccinated COVID-19 patients needing hospitalization due to COVID-19 infection, older age, lower real-time PCR cycle threshold values and a shorter duration between symptom onset and hospital admission were associated with a lack of anti-S SARS-CoV-2 antibodies and poor clinical outcomes of COVID-19 disease [23]. Moreover, published data from the COVAX study revealed that diabetic patients had significantly higher possibilities to have undetectable antibodies on hospital admission due to COVID-19 [24]. 

The beneficial role of the national vaccination campaign in Greece in terms of SARS-CoV-2 cases, ICU admissions and mortality is well-documented [25]. Another recent study by Lytras et al. [24] documented the high and durable effectiveness of COVID-19 vaccination in preventing severe disease and mortality in all age groups, both against Delta and older SARS-CoV-2 variants; however, the study lacks data regarding the Omicron strain. It is estimated that vaccination prevented approximately 19,691 COVID-19 deaths during 2021 in Greece [26].

Several studies reported a reduction in effectiveness of COVID-19 vaccination against variants of the virus compared to the original strain [27,28,29,30]. The accumulation of mutations in these VOCs and others demonstrates the quantifiable risk of antigenic drift and subsequent reduction in vaccine efficacy [31]. Neutralization titers against VOCs were 3-fold lower when analyzing convalescent sera and 3.3-fold and 2.5-fold lower for Pfizer and AstraZeneca vaccinees, respectively [27]. Moreover, Kodera et al. [32] estimated the effectiveness of vaccination at 62.1% (95% CI: 48–66%) compared to the Delta variant. However, the impact of this reduction of the vaccination’s effectiveness in molecular level with waning antibody levels remains unclear. Our study confirms the protective role of vaccination from severe disease caused by mutated virus strains. 

A finding of a great interest is the fact that vaccination modifies common risk factors that had been strongly associated with COVID-19-attributed mortality and morbidity. Underlying comorbidities including hypertension, diabetes, chronic respiratory diseases, cardiac disease and malignancy have been previously associated with severe infection [33,34]. According to the results of several studies, acute and chronic kidney disease, COPD, diabetes, hypertension, cardiovascular disease, cancer, increased D-dimer, male gender, older age, being a current smoker and obesity are clinical risk factors associated with mortality due to COVID-19 [35,36,37]. In univariate analysis of our data, atrial fibrillation, coronary disease and hypertension were associated with 3.5, 4 and 3.5 times increased mortality risk, respectively, compared with patients without these comorbidities. Moreover, diabetic patients were associated with 2.5 times increased possibility of mortality, while patients with chronic respiratory diseases were 2.6 times more likely to decease after the COVID-19 infection. The modification of these possibilities in multivariate analysis indicates the crucial protective role of vaccination in high-risk patients. In line with these observations, national vaccination campaigns have to focus on vulnerable populations that are expected to be benefited the most by vaccination’s protection. Campaigns informing the general population regarding the safety and efficacy of the new COVID-19 vaccines could reinforce the acceptability of vaccination [38,39].

Prior to the appraisal of our results, several limitations should be considered. Data were collected from a database of consecutive patients, eliminating the possibility of selection bias. However, information bias may have occurred due to the retrospective design of the present study, depending on the availability and accuracy of data records. Moreover, missing data regarding current smoking status and the lack of data regarding obesity profile remain two major limitations of our study. Furthermore, data were collected from a single tertiary hospital including only hospitalized patients, and the generalization of our findings is limited. 

## 5. Conclusions

Vaccination against the SARS-CoV-2 virus had a protective effect on terms of mortality and clinical severity of COVID-19 during the fourth wave of the pandemic in Central Greece. Moreover, vaccination modified common risk factors, including hypertension, diabetes, chronic respiratory diseases, cardiac disease and malignancy, that had been strongly associated with COVID-19-attributed mortality. The national vaccination policy has to focus on vulnerable populations that are expected to benefit the most from the vaccine’s protection.

## Figures and Tables

**Figure 1 jpm-12-01423-f001:**
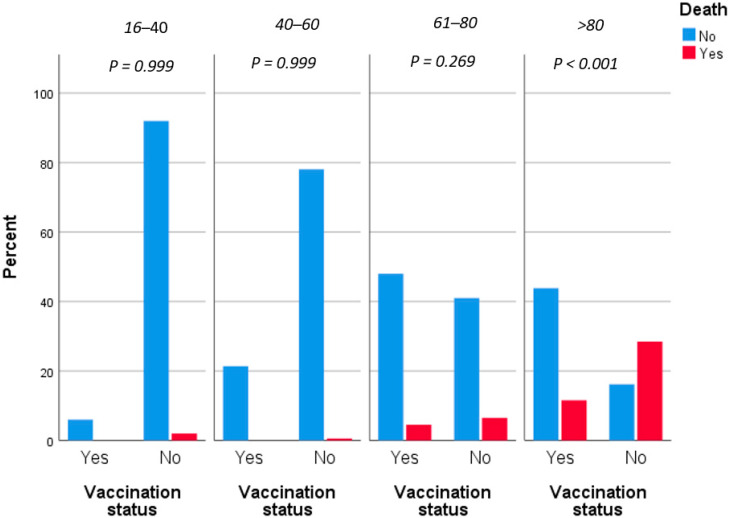
Fatality rates grouped by vaccination status and adjusted by age.

**Figure 2 jpm-12-01423-f002:**
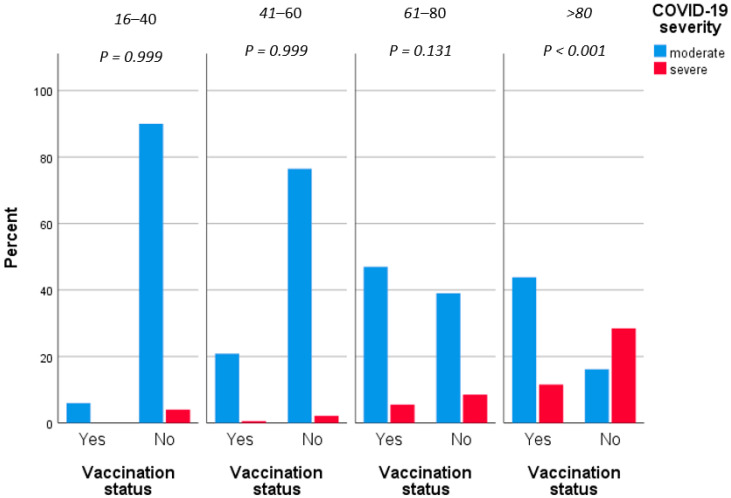
Severity grouped by vaccination status.

**Figure 3 jpm-12-01423-f003:**
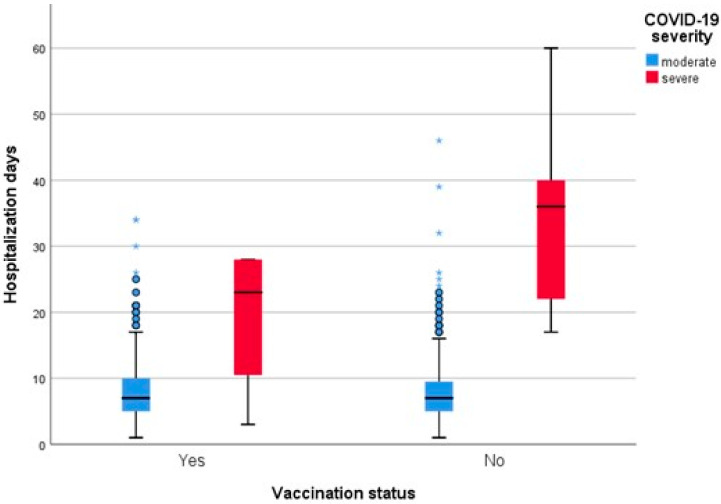
Length of hospitalization grouped by vaccination status and adjusted by clinical severity.

**Table 1 jpm-12-01423-t001:** Assessment of mortality risk due to COVID-19 by univariate and multivariate statistical analysis.

	Mortality	Univariate			Multivariate Binary Logistic Regression	
	Yes (%)	Significance	OR with 95% CI	RR with 95% CI	Significance	aOR with 95% CI
Male gender	45 (13.9)	0.805 (C)	1.09 (0.67–1.78)	1.08 (0.71–1.65)	0.013	2.29 (1.19–4.39)
Age (median, IQR)	Dead: 83 (13) Alive: 61 (26)	<0.001 (M-W)	–		<0.001	1.12 (1.09–1.16)
Smoking *	2 (3.6)	0.521 (F)	0.48 (0.10–2.24)	0.50 (0.11–2.19)	Insufficient data	
Vaccination	24 (10.9)	0.169 (C)	0.70 (0.42–1.17)	0.73 (0.46–1.15)	<0.001	0.20 (0.10–0.39)
Atrial fibrillation	23 (29.9)	<0.001 (C)	3.47 (1.98–6.10)	2.73 (1.79–4.18)	0.500	1.27 (0.63–2.56)
Chronic respiratory fisease	14 (26.4)	0.004 (C)	2.59 (1.33–5.04)	2.17 (1.31–3.60)	0.437	1.40 (0.60–3.24)
Coronary disease	27 (30.7)	<0.001 (C)	3.83 (2.24–6.57)	2.96 (1.97–4.46)	0.549	1.24 (0.62–2.47)
Diabetes type II	23 (24.2)	0.001 (C)	2.50 (1.44–4.32)	2.13 (1.38–3.30)	0.218	1.56 (0.77–2.18)
Malignancy	7 (14.9)	0.768 (C)	1.14 (0.49–2.63)	1.12 (0.54–2.28)	0.803	1.14 (0.42–3.09)
Dyslipidemia	39 (23.4)	<0.001 (C)	2.94 (1.80–4.79)	2.48 (1.65–3.74)	0.090	1.75 (0.92–3.35)
Hypertension	57 (20.2)	<0.001 (C)	3.41 (1.99–5.84)	2.92 (1.80–4.73)	0.692	1.16 (0.56–2.38)
C: Chi-square test; F: Fisher’s exact test M-W: Mann-Whitney U test						

* Insufficient data.

**Table 2 jpm-12-01423-t002:** Assessment of clinical severity of COVID-19 by univariate and multivariate statistical analysis.

	Severe COVID-19 (N = 89)	Univariate			Multivariate Binary Logistic Regression	
		Significance	OR with 95% CI	RR with 95% CI	Significance	aOR with 95% CI
Male gender	51 (15.8)	0.879 (C)	1.04 (0.66–1.64)	1.03 (0.70–1.52)	0.025	1.92 (1.09–3.41)
Age (median, IQR)	Moderate: 61 (13) Severe: 83 (26)	<0.001 (M-W)	–		<0.001	1.09 (1.06–1.11)
Smoking *	5 (8.9)	0.789 (F)	1.16 (0.39–3.46)	1.15 (0.42–3.11)	Insufficient data	
Vaccination	27 (12.3)	0.109 (C)	0.67 (0.41–1.10)	071 (0.47–1.09)	<0.001	0.25 (0.14–0.45)
Atrial fibrillation	23 (29.9)	<0.001 (C)	2.76 (1.59–4.80)	2.24 (1.49–3.37)	0.712	1.13 (0.59–2.19)
Chronic respiratory disease	15 (28.3) 74 (14.3)	0.007 (C)	2.37 (1.24–4.52)	1.98 (1.23–3.19)	0.406	1.39 (0.64–3.04)
Coronary disease	27 (30.7) 62 (12.8)	<0.001 (C)	3.01 (1.78–5.08)	2.39 (1.62–3.53)	0.892	1.05 (0.55–2.00)
Diabetes type II	25 (26.3) 64 (13.4)	0.002 (C)	2.30 (1.36–3.90)	1.96 (1.30–2.94)	0.272	1.44 (0.75–2.78)
Malignancy	11 (23.4) 78 (14.9)	0.123 (C)	1.75 (0.85–3.58)	1.57 (0.90–2.74)	0.142	1.85 (0.81–4.23)
Dyslipidemia	44 (26.3) 45 (11.1)	<0.001 (C)	2.85 (1.80–4.54)	2.37 (1.63–3.44)	0.056	1.76 (0.99–3.12)
Hypertension	63 (22.3) 26 (9.0)	<0.001 (C)	2.91 (1.78–4.75)	2.48 (1.62–3.80)	0.748	1.11 (0.59–2.07)
C: Chi-square test; F: Fisher’s exact test M-W: Mann-Whitney U test						

* Insufficient data.

## Data Availability

Data are available upon reasoning request.

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
