# Peer review of "COVID-19 Severity and Mortality after Vaccination against SARS-CoV-2 in Central Greece"

_jpm, 2022, doi:10.3390/jpm12091423_

Round 1

Author Response

  1. The whole paper contains grammatical mistakes as well as various syntactical inaccuracies.

The authors should improve the language’s quality of the text.

Authors’ answer:  The whole manuscript underwent a grammar revision.

  1. The title contains a redundancy in form of the expression “SARS-CoV-2 Virus”, as the

acronym SARS-CoV-2 already contains the notion of virus. The title itself should therefore

be modified.

Authors’ answer:  Thank you for the well-stated comment. The title was modified accordingly.

  1. The Abstract should be added with a general introduction about the state-of-the-art

knowledge about SARS-CoV-2 and its vaccines, possibly taken from the paper’s

Introduction section.

Authors’ answer:  Thank you for the well-stated comment. The abstract section was modified accordingly.

  1. Introduction section, paragraph three: before talking about the Omicron variant’s impact

on the pandemic’s data, it would be appropriate to define variants of concern and Omicron

itself.

Authors’ answer:  Thank you for the well-stated comment. The introduction section was modified accordingly.

  1. Materials and Methods section, Participants and Study Design subsection: the phrase

“COVID-19 infection was the single common official cause of death for our study

participants, as registered in hospital archives” should be moved to the Results section.

Authors’ answer:  The phrase was moved accordingly.

  1. Materials and Methods section, Participants and Study Design subsection: the last

paragraph of the subsection contains a relevant mistake, stating that there is no sufficient

data about the association of smoking with COVID-19-related mortality. As a matter of

fact, various systematic reviews and meta-analysis exist stating that smoking is a risk factor

for SARS-CoV-2-related negative outcome. Furthermore overweight or obesity condition

are not investigated, but this health condition impact in COVID outcome. It probably

should be considered a limit of the study analysis and should be reported in the discussion.

The following articles may be used as references:

  1. Dessie ZG, Zewotir T. Mortality-related risk factors of COVID-19: a systematic

review and meta-analysis of 42 studies and 423,117 patients. BMC Infect Dis. 2021

Aug 21;21(1):855. doi: 10.1186/s12879-021-06536-3. PMID: 34418980; PMCID:

PMC8380115.

  1. Izcovich A, Ragusa MA, Tortosa F, Lavena Marzio MA, Agnoletti C, Bengolea A,

Ceirano A, Espinosa F, Saavedra E, Sanguine V, Tassara A, Cid C, Catalano HN,

Agarwal A, Foroutan F, Rada G. Prognostic factors for severity and mortality in

patients infected with COVID-19: A systematic review. PLoS One. 2020 Nov

17;15(11):e0241955. doi: 10.1371/journal.pone.0241955. Erratum in: PLoS One.

2022 May 26;17(5):e0269291. PMID: 33201896; PMCID: PMC7671522.

  1. Vimercati L, De Maria L, Quarato M, Caputi A, Gesualdo L, Migliore G, Cavone

D, Sponselli S, Pipoli A, Inchingolo F, Scarano A, Lorusso F, Stefanizzi P, Tafuri

  1. Association between Long COVID and Overweight/Obesity. J Clin Med. 2021

Sep 14;10(18):4143. doi: 10.3390/jcm10184143. PMID: 34575251; PMCID:

PMC8469321.

Authors’ answer:  Thank you for the well-stated comment. In lines 80-83 we state that current smoking has not been associated with COVID-19 in univariate analysis of our data. However, because of the great number of missing data regarding smoking status, our results regarding smoking status have limited value. Furthermore, data regarding patients’ weight were not recorded. In discussion section we have added a paragraph reporting the above-mentioned studies and the limitations of our study.

  1. Results section, paragraph one: “the majority” is not a number, and should therefore be

replaced with the actual number of patients diagnosed with moderate COVID-19

symptoms.

Authors’ answer:  Thank you for the comment. This phrase was modified accordingly.

  1. Results section, paragraph four: the OR for increased age was not provided.

Authors’ answer:  In univariate analysis, for quantitative variables as age, OR cannot be calculated, as stated in table 2.

  1. Discussion section, paragraph six: references 22 to 25 should be briefly discussed in order to compare their data with this study’s.

Authors’ answer:  Thank you for the well-stated comment. The discussion section was modified accordingly and in lines 205-213 the results of the above-mentioned studies were discussed.

  1. Discussion section, paragraph six: the sentences “However, the impact [...] remains

unclear” and “Our study [...] mutated virus’ strains” should be rewritten as they currently

lack meaning, probably due to omissions of a few words.

Authors’ answer:  Thank you for the well-stated comment. These lines were modified accordingly.

  1. Reference 1 is incomplete, as the website’s URL was not fully provided.

Authors’ answer:  Thank you for the comment. The reference URL was provided.

  1. The following articles could be implemented as references:
  2. Bianchi FP, Tafuri S, Migliore G, Vimercati L, Martinelli A, Lobifaro A, Diella G,

Stefanizzi P; Control Room Working Group. BNT162b2 mRNA COVID-19

Vaccine Effectiveness in the Prevention of SARS-CoV-2 Infection and

Symptomatic Disease in Five-Month Follow-Up: A Retrospective Cohort Study.

Vaccines (Basel). 2021 Oct 7;9(10):1143. doi: 10.3390/vaccines9101143. PMID:

34696252; PMCID: PMC8538139.

Authors’ answer:  Thank you for the comment. The above-mentioned reference was added in discussion section.

Reviewer 2 Report

This is an interesting paper focused on  COVID-19 mortality that requires addressing some issues.

 Introduction.

a)     Information on mortality and vaccines’ effectiveness is absent.

b)     The justification is not clear. It is essential to distinguish what is currently known from what the study contributes.

c)     The objective of the study was “to assess COVID-19 attributed morbidity and mortality after the prevalence of new variants of the virus (Delta and Omicron viral strains) and to evaluate the vaccination effect”. Could you please mention what vaccine or vaccines? Was only full vaccination considered? Were the morbidity, mortality, and vaccines’ effectiveness measured only in hospitalized patients? Please specify.

d)     The virus variant type was not measured, consequently, the authors cannot state  “… morbidity and mortality after the prevalence of new variants of the virus (Delta and Omicron viral strains)”.

e)     What does “COVID-19 attributed morbidity” mean? It makes sense to measure COVID-19-attributed mortality, but not morbidity. You collected data on “common morbidities”  (e.g., atrial fibrillation, chronic respiratory disease, diabetes, hypertension), which are potential risk factors for dying from COVID-19, not morbidity attributed to COVID. It is necessary to refine the wording of the objective.

f)      After reading the material and methods section I realized the objective of the study was centered on analyzing the association between clinical, demographics, and mortality due to COVID-19 in hospitalized cases.  Please revise the wording of the objective.

 Material and methods

a)     Was the vaccination scheme measured? The objective states “… to evaluate the vaccination effect”.

Results.

a)     What does “moderate and severe symptomatology” mean? (This should be defined in the material and methods section).

b)  The authors mentioned that 38,7% were fully vaccinated against SARS-CoV-2. But, there is no information on collecting this data in the material and methods section.

c)     The “mortality rate” was calculated at 10.1%. This result seems to be the fatality rate, not the mortality rate. Please revise.

d)   Figures 1 and 2 data are not adjusted by age or other risk factors. Please, at least adjust by age.

e) Figure 3 needs to be adjusted by severity.

Discussion

a)     Authors recognize the results of COVID-19 attributed mortality are biased to patients that required hospitalization.

 Conclusions.

a)     Please remove “morbidity” from the paragraph “vaccination modified ... been strongly associated with COVID-19 attributed mortality and morbidity”.

Author Response

Introduction.

  1. a) Information on mortality and vaccines’ effectiveness is absent.

Authors’ answer:  Thank you for the well-stated comment. Information about vaccination effectiveness were added in introduction section.

  1. b) The justification is not clear. It is essential to distinguish what is currently known from what the study contributes.

Authors’ answer:  Thank you for the well-stated comment. The justification was added.

  1. c) The objective of the study was “to assess COVID-19 attributed morbidity and mortality after the prevalence of new variants of the virus (Delta and Omicron viral strains) and to evaluate the vaccination effect”. Could you please mention what vaccine or vaccines? Was only full vaccination considered? Were the morbidity, mortality, and vaccines’ effectiveness measured only in hospitalized patients? Please specify.

Authors’ answer:  Thank you for the well-stated comment. In results section (lines 109-112) we have stated the vaccinations that were used. Only fully vaccinated patients were included in the present study (line 64). Moreover, only data from hospitalized patients were included in the study as it is mentioned in study design (lines 67-68).

  1. d) The virus variant type was not measured, consequently, the authors cannot state “… morbidity and mortality after the prevalence of new variants of the virus (Delta and Omicron viral strains)”.

Authors’ answer:  Thank you for the well-stated comment. The virus variants prevalence was considered based on epidemiological and seroprevalence studies that were conducted in Greece during the study period.

  1. e) What does “COVID-19 attributed morbidity” mean? It makes sense to measure COVID-19-attributed mortality, but not morbidity. You collected data on “common morbidities” (e.g., atrial fibrillation, chronic respiratory disease, diabetes, hypertension), which are potential risk factors for dying from COVID-19, not morbidity attributed to COVID. It is necessary to refine the wording of the objective.

Authors’ answer:  Thank you for the well-stated comment. Aim and objective was modified accordingly.

  1. f) After reading the material and methods section I realized the objective of the study was centered on analyzing the association between clinical, demographics, and mortality due to COVID-19 in hospitalized cases. Please revise the wording of the objective.

Authors’ answer:  Thank you for the well-stated comment. Aim and objective was modified accordingly.

 Material and methods

  1. a) Was the vaccination scheme measured? The objective states “… to evaluate the vaccination effect”.

Authors’ answer:  Thank you for the well-stated comment. In results section (lines 109-112) we have stated the vaccinations that were used. Only fully vaccinated patients were included in the present study (line 64).

Results.

  1. a) What does “moderate and severe symptomatology” mean? (This should be defined in the material and methods section).

Authors’ answer:  Thank you for the well-stated comment. Clinical severity of COVID-19 was classified by the WHO classification (lines 84-85).

  1. b) The authors mentioned that 38,7% were fully vaccinated against SARS-CoV-2. But, there is no information on collecting this data in the material and methods section.

Authors’ answer:  Vaccination status was recorded from the national vaccination registry (methods -lines 79-80)

  1. c) The “mortality rate” was calculated at 10.1%. This result seems to be the fatality rate, not the mortality rate. Please revise.

Authors’ answer:  Thank you for the well-stated comment. Mortality was replaced with fatality (line 183).

  1. d) Figures 1 and 2 data are not adjusted by age or other risk factors. Please, at least adjust by age.

Authors’ answer:  Figures 1 and 2 were adjusted by age. We chose these age groups according to age distribution (histogram). We wanted these age groups to be timely and population equal.

  1. e) Figure 3 needs to be adjusted by severity.

Authors’ answer:  Thank you for the well-stated comment. Figure 3 was adjusted by severity.

Discussion

  1. a) Authors recognize the results of COVID-19 attributed mortality are biased to patients that required hospitalization.

Authors’ answer:  The present limitation has been added in limitation paragraph in discussion section.

 Conclusions.

  1. a) Please remove “morbidity” from the paragraph “vaccination modified ... been strongly associated with COVID-19 attributed mortality and morbidity”.

Authors’ answer:  Conclusions have been modified accordingly.

Round 2

Reviewer 2 Report

Comments to the paper focused on  COVID-19 severity and mortality after Vaccination Against SARS-CoV- 2 in Central Greece.

General: It would be appropriate and opportune to adjust the title given the content of the results presented.  Please consider removal/replacement of the term "morbidity".

Specific:

-Line 55  It is more accurate to use the term “clinical severity” than “morbidity” (in the manuscript context)

Material and methods

-Line 86. It says: “clinical and demographic variants”. It should say “clinical and demographic characteristics” (or “factors”, but not “variants”).

 Results.

-Line 128. It says: “mortality rate”: It should say “fatality rate” (same as line 198)

-Line 156. It says: “mortality rates”: It should say “fatality rates” (to standardize terms)

 Discussion OK

Conclusions. OK